# Androgen Receptor Gene CAG Repeat Length Varies and Affects Semen Quality in an Ethnic-Specific Fashion in Young Men from Russia

**DOI:** 10.3390/ijms231810594

**Published:** 2022-09-13

**Authors:** Ludmila Osadchuk, Gennady Vasiliev, Maxim Kleshchev, Alexander Osadchuk

**Affiliations:** Department of Human Molecular Genetics, Federal Research Center ‘Institute of Cytology and Genetics’, The Siberian Branch of the Russian Academy of Sciences, Prospekt Lavrentyeva 10, 630090 Novosibirsk, Russia

**Keywords:** genetic-association study, *AR* CAG repeat polymorphism, androgen receptor, semen quality, reproductive hormones, male fertility

## Abstract

Male infertility is a multi-factorial and multi-genetic disorder, and the prevalence of male infertility in the world is estimated at 5–35%. The search for the causes of male infertility allowed for identifying a number of genetic factors including a single X-linked gene of the androgen receptor (*AR*), and some of its alleles are assumed to negatively affect male fertility. Our aim was (1) to study the variability of the length of CAG repeats of the *AR* gene and possible associations in the *AR* CAG genetic variants with semen quality and reproductive hormone levels in a population-based cohort of men and (2) to estimate distributions of *AR* CAG repeat alleles and associations with semen parameters in different ethnic subgroups. The cohort of 1324 young male volunteers of different ethnicities (median age 23.0 years) was recruited from the general population of five cities of the Russian Federation, regardless of their fertility status. Semen quality (sperm concentration, motility and morphology), reproductive hormone levels (testosterone, estradiol, LH, FSH and inhibin B) and trinucleotide (CAG) n repeat polymorphism in exon 1 of the *AR* gene were evaluated. The semen samples were analyzed according to the WHO laboratory manual (WHO, 2010), serum hormones were measured by enzyme immunoassay, and the *AR* CAG repeat length was analyzed by direct sequencing of leukocyte DNA. The median *AR* CAG repeat length in men of our multi-ethnic population was 23 (range 6–39). In the entire study population, a significant difference (*p* ≤ 0.05) was found in the frequency distribution and the mean values for the CAG repeat length between the groups with normal (23.2 ± 3.3) and impaired semen quality (23.9 ± 3.2). Additionally, we demonstrated that the total sperm count, sperm concentration, progressive motility and normal morphology were lower in the category of long CAG repeats (CAG ≥ 25) compared with those in the category of short CAG repeats (CAG ≤ 19); however, hormonal parameters did not differ between the long and short CAG categories, with the exception of estradiol. Significant differences were observed in the *AR* CAG repeat length between the most common ethnic cohorts of Slavs (Caucasians), Buryats (Asians), and Yakuts (Asians). The Buryats and Yakuts had a higher number of CAG repeats than the Slavs (medians: Slavs—23; Buryats—24; Yakuts—25). The range of alleles differed among ethnicities, with the Slavs having the largest range (7–36 repeats, 24 alleles total), the Yakuts having the smallest range (18–32 repeats, 14 alleles total) and the Buryats having the middle range (11–39 repeats, 20 alleles total). The longer CAG repeats were associated with an impaired semen quality within the Slavic (CAG ≥ 25) and Buryat (CAG ≥ 28) groups, but this effect was not found in Yakuts. Hormonal parameters did not differ between the three CAG repeat categories in men of all ethnic groups. This is the largest Russian study of the distribution of *AR* CAG repeats and the search for association between length of *AR* CAG repeat tract and impaired spermatogenesis in men from the general population. Our results confirmed the association of longer CAG repeats with a risk of impaired semen quality, but this association can be modified by ethnic origin. Identification of the number of *AR* CAG repeats can be an effective tool to assess the risk of male subfertility and the control of androgen hormone therapy of reproductive diseases.

## 1. Introduction

Male infertility is a multi-factorial and multi-genetic disorder, and the incidence of male infertility in the world is estimated at 5–35%, which may reflect real differences between populations [1]. Currently known causes of male-factor infertility account for only 60% of cases, and genetic factors account for at least 15% of male infertility [2]. The search of genetic causes of the global decline in male fertility observed in recent decades in many countries is one of the hot spots of reproductive genetics [3,4,5]. One of the causes of male infertility and subfertility is an impairment of hormonal regulation of reproductive processes. Androgens, which are secreted by the Leydig cells of the testes and act through the androgen receptor (AR), regulate a wide range of reproductive and no reproductive processes. They are essential for sexual differentiation, pubertal development, sexual behavior and spermatogenesis. The *AR* gene contains eight exons and seven introns, located on chromosome Xq11-12 [6]. The AR, like other members of the steroid receptor superfamily, has three important basic domains, including a highly conserved central DNA-binding domain connected by a hinge region to a C-terminal ligand-binding domain, and an N-terminal transactivation domain. Two polymorphisms, the CAG and GGN polymorphisms, located in exon 1 of the *AR* gene, code for polyglutamine and polyglycine stretches, respectively. It was found that longer CAG repeats are associated with a decrease in AR transactivation activity, and clinical studies have shown that longer polyglutamine tracts are a factor affecting spermatogenesis and male fertility [5,6,7].

The maximum range of the length of the *AR* gene CAG repeats expands from 6 to 40 trinucleotides; however, ethnic differences in the length of CAG repeats were revealed [8,9,10,11,12,13]. A review of the available studies showed that, in males of the Negroid race, the maximum range of CAG repeats is 13–34, although some populations have longer CAG repeats; in males of the Caucasian race living in different regions of the world, it is 9–31; in males of the Mongoloid race living in East Asia, it is 6–40 triplets [5]. Significant ethnic differences in the frequencies of the *AR* CAG alleles suggest potential ethnic differences in the transcriptional activity of the androgen receptor and, consequently, in the sensitivity of target tissues, including reproductive ones, to androgens.

Quantitative impairment of spermatogenesis accounts for approximately 75% of cases with male factor infertility [2]. The association between the CAG repeat length and an impairment of spermatogenesis or male infertility has been investigated in many studies, but the conclusions were controversial. Within the physiological range, an increased CAG repeat length is assumed to correlate with the reduced androgen sensitivity resulting in an impaired spermatogenesis. Most of the studies have shown an association between the expanded CAG repeats with male infertility and an impairment of spermatogenesis, although this was not true for all studies [6,14,15]. The length of the *AR* CAG repeats had a different effect on male fertility depending on the ethnic composition of the study population: long CAG alleles were associated with the risk of male infertility mainly in Caucasian populations, but such relationships were occasionally observed in some Asian and African populations [15]. Thus, the inconsistency of the results available can be partly explained by ethnic differences; however, significant associations between the CAG repeat length and various forms of pathozoospermia were not always detected in different populations of the same ethnicity or race [15].

For the past decades, several population-based studies addressed the issue of geographic differences in semen quality. A significant amount of data available came from European and Asian countries, and United States of America showing large regional differences in sperm concentration, total count and motility [16,17,18,19]. Our recent studies on young Russian men from the general population have also demonstrated geographic and ethnic differences in semen quality [20,21]. Whether these differences could be caused by genetic, environmental and/or lifestyle-related factors is still an unresolved question. Although the association of *AR* CAG repeat polymorphism with impaired spermatogenesis is not clear and is often contradictory in different populations, it is assumed that knowledge of the population variability of *AR* CAG length may be extremely informative as a basis for understanding ethnic differences in semen quality associated with the androgen action.

The aim of this population-based study was to investigate the variability of the length of *AR* CAG repeats and the association between various *AR* CAG genetic variants with semen quality and reproductive hormone levels in a multi-ethnic male population from Russia. The distribution of *AR* CAG repeat alleles and the association of the *AR* (CAG) n polymorphism with semen quality or reproductive hormone levels were also evaluated in individual ethnic groups.

## 2. Results

### 2.1. Distribution of AR CAG Repeat Alleles and Association with Semen Parameters in the Entire Study Population

Analysis of the *AR* (CAG) n polymorphism in the entire study population revealed 29 alleles with the range of 6–39 repeats (Figure 1A, Table 1). For more information about the AR (CAG) n polymorphism in the entire study population, see the Appendix A. The median CAG repeat length was 23, and the mean ± SD was 23.5 ± 3.3 triplets. The most common CAG allele was 22 (15.6%) in our study population. The distribution of the CAG repeat length was close to normal, although according to the Kolmogorov–Smirnov test, it was significantly different from normal (*p* < 0.01) and was shifted to the right (Figure 1A).

Comparisons of the frequency distribution of CAG repeat length and the mean value for CAG repeat length for the normal and impaired semen quality subgroups are shown in Figure 1B,C and Table 1. According to the Chi-Square test, the frequency distribution of CAG repeats differed significantly (χ^2^_17_ = 36.17, *p* < 0.01) among the subgroups with normal and impaired semen quality (Figure 1). The most common CAG allele was the same in the subgroups with normal and impaired semen quality (22 triplets; 15.4% and 15.9%, respectively), but the mean value for CAG repeat length was significantly longer in men with impaired compared to normal semen quality (Table 1).

The search for associations between the CAG repeat polymorphism of the *AR* gene and reproductive parameters was carried out in two ways. Firstly, the CAG repeat length was estimated depending on semen quality, and secondly, the reproductive data were estimated depending on the category-stratified length of CAG repeats. The CAG repeat length, anthropometric, semen and hormonal characteristics in men with normal and impaired semen quality are presented in Table 1. As mentioned, the CAG repeats length was longer in the impaired compared to normal semen quality subgroup, and as expected, the subgroup with impaired semen quality differed significantly from the normal semen quality subgroup in all semen parameters except for semen volume. The inhibin B level was significantly lower; LH, FSH and estradiol levels were significantly higher (*p* ≤ 0.05) in the participants with impaired compared to normal semen quality. The testosterone level did not differ between these subgroups.

A comparison of semen and hormonal characteristics between various CAG categories in the entire study population showed significant differences (*p* < 0.05) in total sperm count, sperm concentration, progressive motility and normal morphology, and small but statistically significant differences (*p* < 0.05) in the estradiol level (Table 2). The lowest semen parameters and the highest estradiol level have been measured for the participants with long CAG repeats (CAG ≥ 25) in comparison with short CAG repeats (CAG ≤ 19). However, hormonal parameters, as well as semen volume, did not differ between the “long” and “short” CAG categories.

Thus, in our multi-ethnic study population, the frequency distribution of CAG repeats, the CAG repeat length, and the semen parameters differed significantly between the impaired and normal semen quality subgroups; longer *AR* CAG repeats were associated with an impaired semen quality.

### 2.2. Distribution of AR CAG Repeat Alleles and Association with Semen Parameters in Different Ethnic Subgroups

Ethnic composition of our multi-ethnic study population was motley, and only Slavs (Caucasians), Buryats (Asians), and Yakuts (Asians) represented significant ethnic cohorts; thus, we limited our comparative ethnic analysis to three ethnic subgroups, which accounted for 79.4% of our study population. The ethnic distribution was as follows: Slavs 52.5%, Buryats 16.1%, Yakuts 10.8%, other small ethnic groups and descendants from mixed marriages constituted 20.6%.

The numbers of samples genotyped at the *AR* (CAG) n, frequency distributions of the CAG repeat length and mean values for CAG repeat length for the Slavic, Buryat and Yakut ethnic subgroups are summarized in Table 3 and Table 4 and Figure 2. The mean CAG repeat length differed significantly between all the ethnic groups (*p* < 0.001) and was the shortest in the Slavic subgroup and the longest in the Yakut subgroup, the Buryat subgroup ranked middle (Table 3). According to the Chi-Square test, the frequency distribution of GAG repeat length differed significantly among the Slavic, Buryat and Yakut men (Figure 2). In particular, the frequency distribution differed between Slavs and Buryats (χ^2^_15_ = 26.86, *p* < 0.03); Buryats and Yakuts (χ^2^_12_ = 23.74, *p* < 0.02); Yakuts and Slavs (χ^2^_13_ = 66.97, *p* = 0.001). The most common CAG alleles were 22 (16.3%), 22 (12.9%) and 25 (21.6%) triplets in the Slavic, Buryat and Yakut subgroups, respectively (Figure 2). The range of alleles also differed among ethnic groups: Slavs had the largest range (7–36 repeats, total 24 alleles), Yakuts had the smallest range (18–32 repeats, total 14 alleles), and Buryats had the middle range (11–39 repeats, total 20 alleles). Slavs (n = 12, 1.7%) and Buryats (n = 4, 1.9%) had the same percentage of rare alleles (frequency < 0.5%). Yakuts had no rare alleles. For more information about the AR (CAG) n polymorphism in the different ethnic subgroups, see the Appendix A.

The CAG repeat length, semen and hormonal parameters in the Slavic, Buryat and Yakut subgroups with normal and impaired semen quality are presented in Table 4. A significant difference (*p* < 0.05) in the CAG repeat length between men with normal and impaired semen quality was found only in Slavs, remaining lower in the first group compared to the second. These results indicate a higher incidence of long CAG repeats in the Slavic subgroup with impaired semen quality. Additionally, higher LH and FSH and lower inhibin B level, and as expected, lower total sperm count, concentration, motility and normal morphology were demonstrated in men with impaired compared to normal semen quality in the Slavic ethnic group. In the Buryat and Yakut groups, the impaired semen quality was not accompanied by any changes in the CAG repeat length or hormonal levels, with the exception of the LH level in Yakuts. Testosterone and estradiol levels did not differ significantly between the subgroups with different semen quality in all ethnic groups.

The next step in the search for possible associations between the AR CAG repeat polymorphism and reproductive parameters was the stratification of CAG repeat length into three categories: “short”, “medium” and “long” for each ethnic group. The lengths of short and long CAG repeats were calculated taking into account ethnic differences between the ranges and medians of the length of CAG repeats for Slavs, Buryats and Yakuts.

When the short and long CAG allele categories were compared in each ethnic subgroup, the long CAG allele category had significantly (*p* < 0.05) lower sperm progressive motility and normal morphology than the short CAG repeat category in Slavs, and the long CAG allele category had significantly (*p* < 0.05) lower sperm concentration and normal morphology than the short CAG repeat category in Buryats (Table 5). The semen quality was not related to the *AR* CAG repeat length in Yakuts. Hormonal parameters did not differ between the three CAG repeat categories in men of all ethnic groups (Table 5). Therefore, a longer CAG repeat length was associated with a higher risk of impaired spermatogenesis in two of the three ethnic subgroups included in the comparative analysis.

## 3. Discussion

Our population-based cohort study represents the largest Russian study investigating the frequency distribution of CAG repeat length and the effect of the *AR* CAG repeat polymorphism on semen and hormonal parameters in the general population. Our study also includes a comparative analysis of the frequency distribution of CAG repeat length and the association of the *AR* CAG repeat polymorphism with reproductive indicators in three ethnic groups. The data of the first part of our study based on the entire multi-ethnic study population indicate the influence of the CAG polymorphism of the *AR* gene on semen quality and its possible role in the regulation of male fertility. Our results support the opinion that men with longer (CAG) n may have an increased risk of impaired spermatogenesis. More specifically, we revealed a significant difference in the frequency distribution and mean values for the CAG repeat length between subgroups with normal and impaired semen quality. Additionally, we found that all sperm indicators were lower in the category of long CAG repeats compared with those in the category of short CAG repeats.

Currently, the supposed relationship between the CAG repeat length of the *AR* gene and semen parameters including sperm concentration, motility and morphology is the subject of intense discussion, since the available results are contradictory and often unreliable. In an earlier review, it was noted that longer *AR* CAG repeats were associated with infertility and impaired spermatogenesis of different severity [14]. The role of *AR* CAG polymorphism in male infertility was also clarified in a later meta-analysis [23]. The analysis included groups of males of different ethnicity (European, Chinese, Indian, Australian, Israeli), as well as groups with unidentified ethnicity. This meta-analysis provided support for the hypothesis of a close relationship between long CAG repeats and predisposition to reduced male fertility and infertility. Another meta-analysis, which included males of Caucasian, Mongoloid and mixed race, confirmed that infertile males and males with azoospermia have a longer CAG tract compared to fertile males regardless of race [6]. More recent meta-analysis also supported a higher risk of infertility in men with longer or shorter CAG repeats [15]. Later, these conclusions were supported by the studies on many other populations of different ethnic origin [24,25,26].

Statistically significant differences in the length of *AR* CAG repeats in infertile and fertile males are not always detected. The length of *AR* CAG repeats in the Israeli male population of mixed ethnicity (Jews and Arabs) did not differ between the group with idiopathic infertility and pathozoospermia and the control group with a normal semen analysis [27]. No significant correlation between the length of *AR* CAG repeats and the risk of male infertility or defective steroidogenesis was found in the studies of Egyptian [28]; Iranian [29] and Baltic men [30]. An association between longer CAG repeats and defective spermatogenesis could not be found in male populations from India [31], China [32], Turkey [33], and Jordan [34].

In our entire study, there were no hormonal changes between the stratified subgroups with different CAG repeat length, except estradiol. To date, several studies have examined the possible relation of reproductive hormone levels with *AR* CAG number, but results were controversial. A positive correlation between the number of CAG repeats and serum testosterone levels or, in some cases, estradiol levels was shown in large cohorts of healthy European men, which suggested that the *AR* CAG repeat polymorphism explains 6.0–8.5% of interindividual testosterone variations [28,35,36,37]. The authors believed that men with longer AR CAG repeats and higher testosterone and estradiol levels could compensate partly or totally the weaker activity of their AR to maintain an adequate androgenic status. The above conclusions are not consistent with our results, which are in line with other population studies [30,38]. A lack of association or a negative correlation between the CAG repeat polymorphism and reproductive hormones were also published [39,40,41]. The inconsistent data might be a result of multidirectional effects of individual lifestyle factors, for example, diet, obesity, cigarette smoking, alcohol consumption and physical activity on the level of reproductive hormones [42,43].

Currently, genetic testing of the population is becoming relevant, providing information about the prevalence and influence of certain genetic factors on male fertility and allowing predicting and planning preventive, diagnostic and clinical work on the male reproductive health. The assessment of ethnic differences in the length of CAG repeats of the *AR* gene becomes important for understanding the variability of alleles of this gene and differences in sensitivity to androgens. In connection with the above, the second aim of our study was to investigate the distribution of *AR* CAG repeats and possible associations of the length of *AR* CAG repeat tract with spermatogenesis in men from different ethnic groups of Russia.

We have found significant ethnic differences in the frequency distribution of CAG repeat length and confirmed that the association of longer *AR* CAG repeats with the risk of impaired semen quality can be modified by ethnicity. More specifically, the ethnic differences were revealed in the allele distribution of the *AR* (CAG) n polymorphism and the medians of the GAG repeat length between the groups of Slavs (median 23), Buryats (median 24) and Yakuts (median 25). Longer CAG repeats were associated with an impaired semen quality in the Slavic (CAG ≥ 25) and Buryat (CAG ≥ 28) groups, but this effect was not found in Yakuts. In a previous study, we found that Slavic men had a higher semen quality compared to Buryats and Yakuts, apparently due to the higher testicular function in the Caucasian compared to Asian race [21]. A greater number of CAG repeats in Yakuts than in Slavs and Buryats may affect the regulation of transcription of testosterone-dependent genes and, as a result, contribute to the impairment of spermatogenesis and ethnic differences in the semen quality. Our ethnic groups provide an important starting point for a better understanding of the molecular basis underlying ethnic differences in androgen sensitivity and related reproductive outcomes, and emphasize the importance of using different ethnic groups to study associations of other reproductive genes with semen quality and male infertility.

Our ethnic findings confirmed the previously obtained facts about ethnic differences in the (CAG) n repeat polymorphism. It is well documented that the mean CAG tract length in men of African descent possess significantly shorter alleles than in men of non-African populations [44]. Significant ethnic differences in the allele frequencies of the *AR* (CAG) n polymorphism have been revealed between Caucasian, Thai, Afro-Caribbean, Hispanic populations [13]. In the Mediterranean region, the CAG repeat length is high within population variation. The population distribution of short, medium, and long *AR* CAG alleles is remarkably different between South Europeans, North Africans and Sub-Saharan Africans [12]. Population studies have revealed not only genetic differences in the frequencies of alleles of the CAG polymorphism of the *AR* gene between races or ethnicities, but also associated reproductive differences, including male infertility, impaired spermatogenesis and diseases of the reproductive system.

The most interesting result of the current study was the association of longer *AR* CAG alleles with impaired semen quality found only in Slavs and Buryats, while this effect was not detected in Yakuts. Consequently, the number of CAG repeats can have different effects on the male reproductive function depending on ethnic background. It was noted that the association of long *AR* CAG alleles with male infertility or poor semen quality was confidently established for populations of the Caucasian race, but such associations were rare in Asian and African populations, which was confirmed by the conclusions of several meta-analyses [6,15].

The mechanisms demonstrating how long or short *AR* CAG alleles affect spermatogenesis are insufficiently studied. As is well known, testosterone plays a key role in spermatogenesis, but its functions are mediated by the AR. The effect of testosterone on spermatogenesis is carried out mainly through Sertoli cells, which express the androgen receptor and provide output of mature spermatozoa, and through the androgen-dependent and androgen receptor-expressing epididymis, where spermatozoa acquire motility and higher concentration [5,45,46]. Longer *AR* CAG repeat length, reducing the sensitivity of Sertoli cells and epididymis to androgens, can attenuate the physiological effects of androgens on the sperm output. From the information presented above, we could suggest that the receptor’s sensitivity to testosterone is involved in the impairment of semen quality in the Slavic or Buryat carriers of longer *AR* CAG repeats. Additionally, it should keep in mind that mainly intratesticular testosterone plays a major role in spermatogenesis, and the testicular testosterone content is 25–125 times higher than the blood serum [47]. The serum testosterone level does not fully reflect the intratesticular testosterone content and may not always serve as a marker of an impaired spermatogenesis.

In the Yakut group with longer (CAG) n repeats in the *AR* gene, the compromised transcriptional activity of AR was not connected with signals of impaired spermatogenesis or with hormonal levels. Although the size of this group was not big enough for the final interpretation of the results, the negative results obtained can confirm that long (CAG) n repeats do not always impair the semen quality. In addition, in other studies, no association between the expansion of *AR* CAG repeats and impaired spermatogenesis was found, including Iranian, Swedish, German, Dutch, Belgian, Danish, and Japanese populations [6,14,15,48]. The length of (CAG)n repeats or LH, FSH and free testosterone levels did not differ in four groups of German males with normal semen analysis; proven fertility; infertile with azoospermia or with familial infertility, but these groups differed significantly in sperm concentration [49]. There is no clear explanation for such results, although a possible hypothesis related to the modulation of AR transcriptional activity could be considered. The transcriptional activity of androgen-bound AR is modulated by specific proteins known as coregulators (coactivators or corepressors), which can change the AR ability to transactivate the target genes via chromatin remodeling and histone modification [50]. The assumption that in Yakuts, a decrease in the harmful effects of longer (CAG) n repeats on spermatogenesis may reflect the effect of putative coactivators on AR requires further study. It is also possible that in this ethnic group, unlike others, there are other specific polymorphisms or genetic variants, which overlap this negative effect on spermatogenesis. The role of CAG repeats in the endocrine regulation of spermatogenesis is probably more complex than previously thought; thus, a definite explanation for the results obtained can be given by conducting more thorough future clinical studies.

It is believed that genetic testing subfertile or infertile males for the (CAG) n repeat polymorphism has diagnostic and prognostic value, since it allows for assessing the risk of androgen-dependent disorders, an effectivity of testosterone therapies and the likelihood of the development of reproductive pathology in offspring. Some authors acknowledge that genotyping of the *AR* CAG repeat polymorphism currently has low diagnostic value due to the lack of clearly defined thresholds assessing the risk of male infertility. A question of choosing the threshold repeat length for diagnostic purposes is of fundamental importance. In most cases, the division into “short”, “medium” and “long” CAG repeat length categories is carried out empirically or based on the results of previously published studies. For example, the threshold value of the length of *AR* CAG repeats as 21 triplets is given in the meta-analysis [15], based on facts that longer alleles are associated with an increased risk of infertility. In another meta-analysis [51], it was proposed to stratify the number of *AR* CAG repeats into three categories, taking the number of CAG repeats equal to 22–23 triplets as a reference limit of the norm as the highest AR activity. The repeat lengths less than 22 or more than 23 led to an increased risk of infertility [51]. A number of authors suggest other threshold values; for example, it is proposed to assign the repeats with the length more than 26 triplets to a group of conditionally long CAG alleles in the Russian male population [52]. In another publication [53], the interval of 20–25 repeats was selected as a criterion for the conditionally normal number of *AR* CAG repeats for Russian males. In our study population, the (CAG) n length ranges between 6 and 39 repeats, with a median value that varies according to the ethnicity and longer CAG repeats increased the risk for an impaired semen quality only in men of two ethnicities from three. Based on these data, it is impossible to find a threshold value of CAG repeats, above which the risk of poor spermatogenesis increases. Thus, there are disagreements in determining the threshold values of *AR* (CAG) n length, which, apparently, need to be eliminated, at least for large ethnic groups, before introducing this polymorphism into the diagnostic protocol.

In our study, there are few limitations associated with effects of *AR* CAG repeat alleles on semen parameters. Firstly, more research is needed to evaluate the effects of CAG repeats in the Yakut ethnic group to confirm or reject our pilot observations. Our multi-ethnic research may also serve as a starting point for further studies, taken into account a wide diversity of nationalities in Russia. Secondly, some studies suggest that GGN repeats in the *AR* gene or specific combinations of CAG and GGN triplets can modulate the AR function [30,39,40]. Additional studies are required to identify a role of independent or combinatory effects of CAG and GGN repeat lengths in spermatogenesis of Russian men. Thirdly, it is assumed that long-term research projects will be aimed at whole-exome sequencing in representatives of various ethnic groups from Russia, providing a search of pathogenic ethno-specific genetic variants, significant associations of molecular genetic markers with the parameters of impaired spermatogenesis and new genetic loci involved in the control of spermatogenesis.

## 4. Materials and Methods

### 4.1. Subjects

For the present study, 1324 males were recruited from five Russian cities: Archangelsk, Novosibirsk, Kemerovo, Ulan-Ude, Yakutsk, and all samples were collected during a 5-year period (2009–2014). The city of Archangelsk is located in European North of Russia within the circumpolar area, the cities of Novosibirsk and Kemerovo in Western Siberia; all three cities have a predominantly Slavic population (approximately 90–95%). The cities of Ulan-Ude and Yakutsk are located in Eastern Siberia, and Yakutsk is located near the Arctic Circle. Buryats make up 32% of the total population of Ulan-Ude, and Yakuts make up 43% of the total population of Yakutsk.

In all cities, the study design and standardized recruitment protocol were the same, which were described earlier in more detail [20,21]. Briefly, the study included male volunteers from the general population regardless of their fertility status. All participants were born or lived for at least 3–5 years in the cities prior to the study. Inclusion criteria for participation in the study were absence of acute general diseases or chronic illness in an acute phase and genial tract infections. Each participant filled in a standardized questionnaire including information on age, place of born, self-identified nationality, family status, some lifestyle characteristics. The data of participants were stored anonymously.

All the participants were examined by the same experienced andrologist and the results of examination were recorded. During the examination, some current urogenital disorders were diagnosed. Body weight (kg) and height were measured in all participants. Body mass index (BMI, kg/m^2^) was calculated. Testicular volume (mL) was estimated by a Prader orchidometer and was presented as bitesticular volume (paired testicular volume). Age was calculated as the difference between year of attendance in study and self-reported year of birth.

Participants provided both blood and semen sample. A fasting blood sample from each participant was drawn from cubital vein in the morning before the semen sample was collected. The serum samples were stored at −40 °C until the hormonal analysis. Each participant was asked about necessity of sexual abstinence for 2–7 days before the examination. Semen samples were collected by masturbation into disposable sterile plastic containers in the special privacy room close to the laboratory. All study subjects were voluntaries and did not receive any financial compensation.

Our study population was multi-ethnic and consisted of men of 20 ethnicities and descendants from ethnic mixed marriages living in five Russian cities. To investigate ethnic differences in the *AR* CAG allele distribution and associations between the CAG repeat polymorphism of the *AR* gene and reproductive parameters, the three most numerous ethnic groups were selected from our multi-ethnic study population. They were Slavs (n = 697), Buryats (n = 208) and Yakuts (n = 134). Participants were stratified into the ethnic groups according to information obtained from the self-reporting questionnaires, taking into account self-identified ethnicity and ethnicities of their mothers, fathers and grandparents. The participant of the Slavic, Buryat or Yakut ethnicity was eligible if the ethnicity of himself, his mother, father and both grandparents was the same. Our research cohort consisted of Slavs living in all five cities, and Buryats and Yakuts living compactly in the cities of Ulan-Ude and Yakutsk, respectively.

Differences in semen parameters between these ethnic groups were presented elsewhere with higher semen quality among Slavs, average among Buryats and lowest in Yakuts [21]. There were no significant differences in the CAG repeat length between the Slavic groups living in all five cities (mean ± SD: 23.2 ± 2.7 for Archangelsk, 22.9 ± 3.1 for Novosibirsk, 23.1 ± 3.2 for Kemerovo, 23.4 ± 3.1 for Ulan-Ude and 23.1 ± 3.4 for Yakutsk, *p* = 0.69).

### 4.2. Semen Analysis

The semen samples were analyzed for semen volume (ml), sperm concentration (×10^6^/mL), sperm motility and normal morphology (percentage) according to the WHO laboratory manual for the examination and processing of human semen [22]. Semen analysis was described earlier elsewhere in more detail [20,21]. Sperm concentration was assessed using the Goryaev’s hemocytometer under light microscope (magnification ×400). Total sperm count was then calculated by multiplying the individual’s sperm concentration by the semen volume. Percentage of spermatozoa with progressive motility was estimated in native ejaculate using an automatic sperm analyzer SFA-500 (Biola, Moscow, Russia). The evaluation principle is based on the measurement of optical density fluctuations in the native ejaculate because of the movement of spermatozoa through an optical channel illuminated by a laser beam. Optical fluctuations are registered by a photodetector; the number of sperm with rapid progressive motility (velocity ≥ 25 µm/s, the WHO class A) and with slow progressive motility (velocity 5–25 µm/s, the WHO class B) was calculated by special software. The sperm motility measurements were carried out three times for each sample, and mean value was calculated. To assess sperm morphology, ejaculate smears were prepared, fixed by methanol and stained by using commercially available kits Diff-Quick (Abris plus, Saint Petersburg, Russia) according to the manufacturer manual. Two hundred spermatozoa were examined for morphology with an optical microscope (Axio Skop.A1, Carl Zeiss, Jena, Germany) at ×1000 magnification with oil-immersion, and the sperm anomalies were listed according to the WHO guidelines [22]. Sperm morphology evaluations were performed in duplicates in random and blinded order, and we report here the percentage of sperm scored as morphologically normal (percentage).

### 4.3. Hormone Assays

Serum hormone concentrations were determined by enzyme immunoassay using commercially available kits “Steroid IFA-Testosterone-01”, “Gonadotropin IFA-LH”, “Gonadotropin IFA-FSH” (Alkor Bio, St. Petersburg, Russia), “Estradiol-IFA” (Xema Medica, Moscow, Russia) and “Inhibin B Gen II ELISA” (Beckman Coulter, Pasadena, CA, USA). The ranges of evaluated concentrations for total testosterone (T), estradiol (E_2_), follicle-stimulating hormone (FSH), luteinizing hormone (LH), and inhibin B (InhB) were 0.2–50 nmol/L, 0.1–20 nmol/L, 2.0–100 mME/mL, 20–90 mME/mL, and 12–105 pg/mL, respectively. The sensitivities for T, E_2_, FSH, LH, InhB were 0.2 mmol/L, 0.025 nmol/L, 0.25 mME/mL, 0.25 mME/mL, 2.6 pg/mL, respectively. The intra- and interassay coefficients of variation were as follows: for T < 8.0%; E_2_ < 8.0%; FSH < 8.0%; LH < 8.0%, InhB ≤ 6.8%.

### 4.4. Genotyping

For genotyping of the *AR* CAG repeats, genomic DNA was extracted from peripheral blood leukocytes using the common phenol-chloroform method. Genotyping of the *AR* CAG repeats was performed using fragment analysis PCR and capillary electrophoresis on a “Nanophor-05” sequencer (Syntol, Moscow, Russia). We obtained *AR* (CAG) n genotypes for 1313 of 1324 men. The method allows for determining the relative length of the product in relation to the length standard and is based on the separation of DNA into fractions by molecular weight. Among the azoospermic participants (n = 31), samples with Y-chromosome deletions as known genetic causes of infertility have been excluded. The evaluation principle of the Y-chromosome deletions is available in a Appendix A.

Ten nanograms of DNA were amplified in 15 µL of reaction mixture containing 2.5 pmol each of fluorescently labeled forward, non-labeled reverse primer and RT-PCT master mix (Syntol, Russia). The primer sequences were as follows: forward, 5′-(FAM)-TCCAGAATCTGTTCCAGAGCGTGC-3′ and reverse, 5′-GCTGTGAAGGTTGCTGTTCCTCAT-3′. Each cycle included a denaturation step at 95 °C for 15 s, a primer-annealing step at 63 °C for 15 s and a primer extension step at 72 °C for 15 s with a subsequent of initial denaturation step at 95 °C for 2 min and a final extension step at 72 °C for 7 min. The amplified samples of 0.7 µL each were mixed with 10 µL of formamide and the GeneScan™ 500 LIZ^®^ Size Standard (dilution 1:100). The sizes of fragments were determined by using a “Nanophor-05” sequencer and Fragment software (Syntol, Russia). The number of CAG repeats was calculated using an allelic ladder of marker fragments, which consisted of eight fragments of different lengths and were used as an internal standard for calculations (lengths of CAG repeats were 12, 19, 23, 25, 27, 29, 33 triplet), which were confirmed by Sanger sequencing.

### 4.5. Statistical Analysis

Statistical analysis was performed using the statistical package “STATISTICA” (version 8.0). The results are presented as mean (SD) as well as median with 5th and 95th percentiles. The Kolmogorov–Smirnov test for normality estimated that the values of the studied anthropometrical, semen and hormonal parameters, as well as bitesticular volume were not normally distributed. The Kruskal–Wallis ANOVA test for comparing multiple independent groups was carried out to find the differences in the CAG repeat length, semen and hormonal parameters among different semen quality or ethnic groups. χ^2^ test was used to compare frequency distributions of the CAG repeat length among different semen quality or ethnic groups. A *p* value < 0.05 was regarded as statistically significant.

The first way to identify associations of the CAG repeat length with semen and hormonal parameters was as follows: the participants of the entire study population or each ethnic group were divided into two semen quality subgroups using the WHO reference limits for sperm concentration, motility and morphology [22]. Men were categorized into the normal semen quality subgroup if sperm concentration was ≥15 mill./mL, progressive motility ≥32%, normal morphology ≥4% and the impaired semen quality subgroup if one or more of these indicators were lower the WHO reference limits.

The second way to identify associations of the CAG repeat length with semen and hormonal parameters was as follows: the participants of the entire study population or each ethnic group were stratified into three CAG categories based on the CAG range restriction with a frequency below 5% for short or long CAG repeat length. The rest of the CAG repeat range represented the medium CAG repeat category. The categorization of the CAG repeat length for the entire study population is presented in Table 2. Similarly, the participants in each ethnic group were also stratified into three CAG categories based on the CAG ethnic range restriction with a lower frequency of 5% for short or long CAG repeat length. The rest of the ethnic repeat range represented the medium CAG repeat category. The categorization of the CAG repeat length for each ethnic group is presented in Table 5.

## 5. Conclusions

This study highlights the influence of the CAG repeat polymorphism in the *AR* gene on spermatogenesis and its role in the regulation of fertility in young men from Russia. Our results support the opinion that men with longer (CAG) n may have an increased risk of impaired spermatogenesis. In particular, the CAG repeat length in the *AR* gene can affect semen quality in an ethnic-specific fashion. However, more studies are required to understand a role of the CAG repeat polymorphism in different ethnicities, specifically a range of allele variants in clinic aspects should considered. Additional epidemiological studies are also needed to validate these findings on the AR function and its correlation with spermatogenesis and male fertility for other ethnic groups.

## Figures and Tables

**Figure 1 ijms-23-10594-f001:**
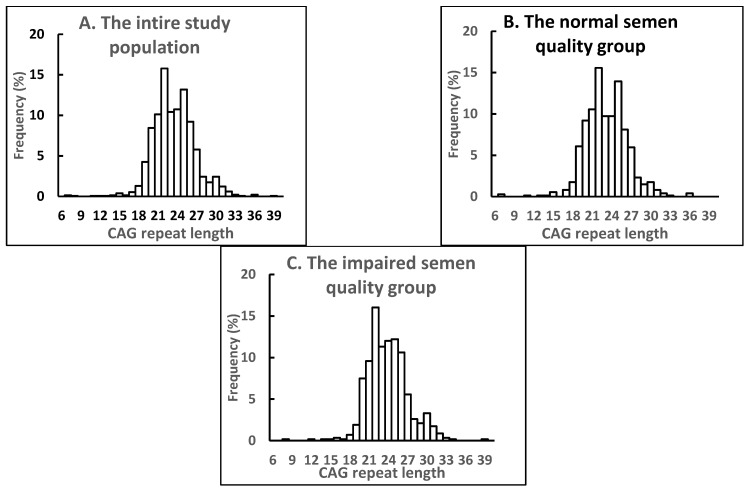
Frequency distribution of CAG repeat length in men of the entire study population (**A**), and the subgroups with normal (**B**) and impaired semen quality (**C**). Values are given as percentage of cases in relation to the respective number of men in each group.

**Figure 2 ijms-23-10594-f002:**
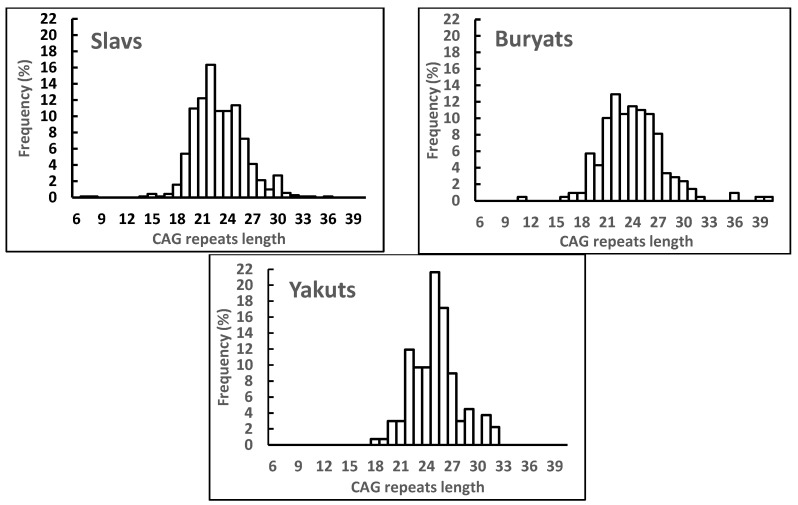
Frequency distribution of the CAG repeat length in the Slavic, Buryat and Yakut ethnic subgroups. Values are given as percentage of cases in relation to the respective number of men in each ethnic group.

**Table 1 ijms-23-10594-t001:** Anthropometric, semen and hormonal characteristics of the entire Russian study population and subgroups with normal and impaired semen quality.

Parameter	The Entire Russian Study Population (n = 1324)		Normal Semen Quality (n = 747)		Impaired Semen Quality (n = 577)	
	Mean (SD)	Median (5–95)	Mean (SD)	Median (5–95)	Mean (SD)	Median (5–95)
Age (years)	25.4 (7.1)	23.0 (18–39)	25.2 (7.0)	23 (18–39)	25.5 (7.4)	23 (18–40)
Weight (kg)	75.6 (14.0)	73.6 (56.5–102.0)	75.7 (13.9)	74.0 (57.0–102.0)	75.4 (14.3)	74.0 (55.6–102.0)
Height (cm)	177.1 (7.1)	177 (165–189)	177.5 (6.8)	177.8 (166.7–188.0)	176.6 (7.5) *	176 (165–190)
BMI (kg/m^2^)	24.1 (4.0)	23.5 (18.7–31.4)	24.0 (4.0)	23.4 (18.5–31.3)	24.1 (4.1)	23.5 (18.8–31.4)
BTV (mL)	39.0 (9.0)	40 (25–54)	40.5 (8.6)	40 (28–56)	37.1 (9.1) *	36 (23–50)
Semen volume (mL)	3.5 (1.6)	3.3 (1.3–6.4)	3.5 (1.6)	3.3 (1.4–6.2)	3.5 (1.6)	3.3 (1.3–6.5)
Total sperm count (×10^6^/ejaculate)	169.7 (170.8)	129.9 (7.8–441.6)	237.4 (189.6)	200.3 (61.4–488.6)	82.5 (83.4) *	57.3 (0.0–246.3)
Sperm concentration (×10^6^/mL)	49.3 (39.8)	40.3 (3.4–129.5)	69.5 (39.7)	58.2 (25.7–147.4)	23.1 (19.4) *	18.8 (0–63.6)
Progressive motility (%)	43.2 (26.7)	40.7 (3.1–89.0)	61.4 (18.1)	60.5 (35.3–92.0)	18.5 (13.0) *	17.7 (1.3–41.8)
Normal morphology (%)	6.6 (3.1)	6.5 (2.0–11.8)	8.2 (2.5)	8.0 (4.8–13.0)	4.4 (2.3) *	4.0 (1.1–8.5)
LH (IU/L)	3.6 (1.6)	3.3 (1.5–6.4)	3.4 (1.4)	3.2 (1.5–6.0)	3.9 (1.8) *	3.6 (1.6–7.1)
FSH (IU/L)	4.2 (3.1)	3.6 (1.5–8.7)	3.8 (2.0)	3.5 (1.5–7.3)	4.8 (4.0) *	4.0 (1.5–10.3)
Testosterone (nmol/L)	20.7 (7.6)	19.7 (10.6–34.2)	20.9 (7.4)	20.1 (10.6–33.4)	20.5 (7.9)	19.0 (10.5–34.7)
Estradiol (nmol/L)	0.21 (0.07)	0.20 (0.12–0.32)	0.20 (0.06)	0.19 (0.11–0.32)	0.21 (0.08) *	0.20 (0.12–0.31)
Inhibin B (pg/mL)	176.1 (66.5)	169.9 (76.1–295.3)	184.9 (64.8)	173.3 (94.2–307.0)	164.9 (67.1) *	161.5 (55.3–277.6)
CAG repeat length	23.5 (3.3)	23 (19–29)	23.2 (3.3)	23 (19–29)	23.9 (3.2) *	24 (20–30)

Abbreviations: SD, standard deviation; (5–95): 5–95th percentile; BMI, body mass index; BTV, bitesticular volume (paired testicular volume); LH, luteinizing hormone; FSH, follicle-stimulating hormone. The stratification of men by different semen quality groups was carried out using the WHO reference limits for sperm concentration, motility and morphology [22]. Men were categorized into the normal semen quality group if sperm concentration was ≥15 mill./mL, progressive motility ≥32%, and normal morphology ≥4%, and into the impaired semen quality group if one or more of these indicators were below the WHO reference limits. The group with impaired sperm quality included azoospermic participants. * differences between the normal and impaired semen quality group are significant (*p* < 0.05).

**Table 2 ijms-23-10594-t002:** Semen and hormonal characteristics of men stratified on various CAG categories in the entire Russian study population.

Parameter	CAG Category (n = 1313)		
	CAG ≤ 19 (Short, n = 95)	20 ≤ CAG ≤ 24 (Medium, n = 1023)	CAG ≥ 25 (Long, n = 195)
Semen volume (mL)	3.6 (1.6)	3.5 (1.6)	3.5 (1.6)
Total sperm count (×10^6^/ejaculate)	188.7 (131.5) ^a^	170.4 (167.7) ^ab^	158.6 (203.4) ^b^
Sperm concentration (×10^6^/mL)	55.7 (40.1) ^a^	49.7 (39.6) ^ab^	44.8 (41.0) ^b^
Progressive motility (%)	51.3 (23.2) ^a^	43.1 (26.8) ^b^	39.9 (27.0) ^b^
Normal morphology (%)	7.9 (2.6) ^a^	6.5 (3.0) ^b^	6.4 (3.1) ^b^
LH (IU/L)	3.7 (1.6)	3.6 (1.6)	3.8 (1.7)
FSH (IU/L)	4.7 (3.1)	4.1 (3.9)	4.4 (3.3)
Testosterone (nmol/L)	19.1 (6.24)	20.8 (7.7)	21.3 (7.9)
Estradiol (nmol/L)	0.19 (0.05) ^a^	0.21 (0.08) ^ab^	0.21 (0.06) ^b^
Inhibin B (pg/mL)	165.9 (64.2)	178.5 (66.5)	169.1 (68.0)

The data are presented as mean (SD). Abbreviations: LH, luteinizing hormone; FSH, follicle-stimulating hormone. The entire study population was divided into three parts according to the CAG repeat length categories: “short”, “medium” and “long” based on the calculated numbers of CAG repeats. ^a,b^ comparisons with various superscript indicators within the variable are significant (*p* < 0.05).

**Table 3 ijms-23-10594-t003:** CAG repeat length in male subgroups of different ethnicity.

		CAG Repeat Length	
Ethnicity	n	Mean (SD)	Median (5–95)
Slavs	697	23.0 ± 3.1 ^a^	23 (19–29)
Buryats	208	24.0 ± 3.5 ^b^	24 (19–30)
Yakuts	134	25.0 ± 2.7 ^c^	25 (21–31)

Abbreviations: SD, standard deviation; (5–95): 5–95th percentile. ^a–c^ comparisons with different superscripts are significant (*p* < 0.05).

**Table 4 ijms-23-10594-t004:** Comparison of CAG repeat length, semen and hormonal parameters among participants with normal and impaired semen quality within the Slavic, Buryat and Yakut ethnic groups.

Parameter	Slavs (n = 697)		Buryats (n = 208)		Jakuts (n = 134)	
	Normal Semen Quality (n = 399)	Impaired Semen Quality (n = 298)	Normal Semen Quality (n = 130)	Impaired Semen Quality (n = 78)	Normal Semen Quality (n = 59)	Impaired Semen Quality (n = 75)
Semen volume (mL)	3.6 (1.7)	3.7 (1.8)	3.3 (1.4)	3.1 (1.2)	3.1 (1.3)	3.1 (1.2)
Total sperm count (×10^6^/ejaculate)	262.7 (199.8)	84.2 (85.8) *	186.2 (135.3)	63.2 (54.8) *	152.4 (82.7)	51.2 (50.6) *
Sperm concentration (×10^6^/mL)	76.4 (42.1)	22.1 (18.8) *	60.0 (37.4)	20.2 (14.2) *	52.2 (24.5)	17.1 (13.9) *
Progressive motility (%)	61.2 (18.3)	15.6 (10.3) *	63.1 (19.7)	17.6 (10.8) *	54.7 (15.4)	18.1 (11.0) *
Normal morphology (%)	8.4 (2.8)	4.6 (2.4) *	7.9 (2.4)	5.0 (2.3) *	6.2 (1.9)	3.4 (1.9) *
LH (IU/L)	3.3 (1.4)	3.8 (1.7) *	3.9 (1.5)	4.1 (1.8)	3.3 (1.4)	3.8 (1.7) *
FSH (IU/L)	3.5 (2.0)	4.1 (3.6) ^+^	4.3 (2.0)	5.4 (3.9)	4.6 (1.7)	5.8 (4.0)
Testosterone (nmol/L)	21.2 (7.5)	21.0 (8.1)	19.0 (5.9)	18.8 (6.8)	20.9 (6.8)	20.5 (7.9)
Estradiol (nmol/L)	0.19 (0.07)	0.20 (0.09)	0.22 (0.06)	0.24 (0.10)	0.22 (0.05)	0.23 (0.05)
Inhibin B (pg/mL)	195.8 (65.4)	173.4 (68.3) *	152.5 (61.3)	137.7 (65.8)	167.1 (47.6)	154.0 (53.9)
CAG repeats length	22.8 (2.9)	23.3 (3.3) *	23.6 (3.4)	24.6 (3.7)	25.2 (2.8)	24.7 (2.6)

The data are presented as mean (SD). Abbreviations: LH, luteinizing hormone; FSH, follicle-stimulating hormone. In each ethnic group, the stratification of men by different semen quality groups was carried out using the WHO reference limits for sperm concentration, motility and morphology [22]. Men were categorized into the normal semen quality group if sperm concentration was ≥15 mill./mL, progressive motility ≥32%, and normal morphology ≥4%, and into the impaired semen quality group if one or more of these indicators were below the WHO reference limits. The group with impaired sperm quality included azoospermic participants. * difference between the normal and impaired semen quality group is significant (*p* < 0.05); ^+^ difference between the normal and impaired semen quality group is close to significant (*p* = 0.057).

**Table 5 ijms-23-10594-t005:** Comparison of semen and hormonal parameters between different CAG categories in the Slavic, Buryat and Yakut ethnic groups.

Slavs (n = 697)			
Parameter	CAG Category		
	CAG ≤ 19 (Short, n = 59)	20 ≤ CAG ≤ 24 (Medium, n = 428)	CAG ≥ 25 (Long, n = 210)
Semen volume (mL)	3.7 (1.8)	3.7 (1.7)	3.7 (1.6)
Total sperm count (×10^6^/ejaculate)	200.4 (136.0)	184.6 (149.9)	196.7 (251.8)
Sperm concentration (×10^6^/mL)	58.9 (44.1)	53.6 (42.2)	54.0 (47.0)
Progressive motility (%)	50.4 (23.4) ^a^	43.5 (27.6) ^ab^	40.8 (27.3) ^b^
Normal morphology (%)	7.7 (2.8) ^a^	7.0 (3.2) ^ab^	6.6 (3.3) ^b^
LH (IU/L)	3.4 (1.4)	3.5 (1.6)	3.7 (1.5)
FSH (IU/L)	4.4 (3.1)	3.7 (2.8)	3.6 (1.9)
Testosterone (nmol/L)	19.4 (6.4)	21.3 (7.8)	21.7 (7.8)
Estradiol (nmol/L)	0.19 (0.06)	0.20 (0.08)	0.19 (0.06)
Inhibin B (pg/mL)	169.2 (65.2)	188.9 (67.0)	183.2 (72.7)
**Buryats (n = 208)**			
**Parameter**	**CAG Category**		
	**CAG ≤ 20** **(short, n = 27)**	**21 ≤ CAG ≤ 27** **(medium, n = 156)**	**CAG ≥ 28** **(long, n = 25)**
Semen volume (mL)	3.0 (1.1)	3.2 (1.4)	3.2 (1.3)
Total sperm count (×10^6^/ejaculate)	133.9 (97.3)	146.0 (133.0)	95.3 (91.0)
Sperm concentration (×10^6^/mL)	46.2 (28.7) ^a^	47.1 (38.1) ^ab^	30.1 (27.3) ^b^
Progressive motility (%)	52.4 (21.6) ^+^	46.6 (28.1)	36.1 (28.7) ^+^
Normal morphology (%)	7.9 (2.6) ^a^	6.8 (2.7) ^ab^	6.3 (3.1) ^b^
LH (IU/L)	4.4 (1.8)	3.8 (1.5)	4.2 (2.0)
FSH (IU/L)	5.6 (3.4)	4.4 (2.2)	5.5 (5.1)
Testosterone (nmol/L)	17.7 (4.9)	19.0 (6.2)	19.1 (5.7)
Estradiol (nmol/L)	0.21 (0.04)	0.23 (0.08)	0.25 (0.10)
Inhibin B (pg/mL)	144.0 (65.6)	149.2 (59.9)	134.5 (79.4)
**Yakuts (n = 134)**			
**Parameter**	**CAG Category**		
	**CAG ≤ 21** **(short, n = 10)**	**22 ≤ CAG ≤ 27** **(medium, n = 94)**	**CAG ≥ 28** **(long, n = 30)**
Semen volume (mL)	2.5 (0.7)	3.1 (1.3)	3.2 (1.4)
Total sperm count (×10^6^/ejaculate)	65.5 (62.1)	102.0 (74.4)	110.2 (117.4)
Sperm concentration (×10^6^/mL)	25.7 (24.0)	35.7 (27.1)	33.2 (25.3)
Progressive motility (%)	29.0 (18.7)	36.8 (23.6)	38.3 (21.4)
Normal morphology (%)	5.4 (2.0)	4.7 (2.4)	4.8 (2.3)
LH (IU/L)	3.1 (1.2)	3.6 (1.6)	3.8 (1.6)
FSH (IU/L)	4.6 (1.6)	5.2 (3.2)	5.5 (3.5)
Testosterone (nmol/L)	20.6 (6.2)	20.0 (6.7)	23.0 (9.2)
Estradiol (nmol/L)	0.22 (0.06)	0.22 (0.05)	0.23 (0.06)
Inhibin B (pg/mL)	162.1 (53.7)	156.7 (48.3)	168.7 (59.2)

The data are presented as mean (SD). Abbreviations: LH, luteinizing hormone; FSH, follicle-stimulating hormone. Each ethnic subgroup was stratified into three parts according to the CAG repeat length categories: “short”, “medium” and “long” based on the calculated numbers of CAG repeats. ^a,b^ comparisons with various superscript indicators within the variable are significant (*p* <0.05); ^+^—difference between the different CAG repeat length categories is close to significant (*p* = 0.085).

## Data Availability

Restrictions apply to the availability of some data of this study to preserve patient confidentiality. The corresponding author will on request detail the restrictions and other conditions under which access to some data may be provided.

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
