# Peer review of "Androgen Receptor Gene CAG Repeat Length Varies and Affects Semen Quality in an Ethnic-Specific Fashion in Young Men from Russia"

_ijms, 2022, doi:10.3390/ijms231810594_

Round 1

Reviewer 1 Report

The study is dedicated to the actual problem of andrology - genetic basis of male infertility. However, there are a number of comments to this article:

1. Line 4, title: “...in Russian men from the general population” – if the samples are collected correctly, the words about the general population are unnecessary; if the samples are not representative, the study was not done correctly. So, it is recommended to remove the words about the general population.

2. To the entire text: the name of the human AR gene should be written in italic in accordance with the international HUGO nomenclature.

3. Line 13-14: “…the androgen receptor (AR) that is assumed to negatively affect male fertility” – not the androgen receptor itself negatively affects fertility, but only some of its alleles.

4. Line 18: “…young Russian men of different ethnicities…” – Russians cannot be of different nationalities; they are a separate ethnic group. Perhaps, the authors mean they collected material from patients (citizens of the Russian Federation) in 5 different regions with significantly different ethnic composition as described in the sample collection section. So, it is better to refer they as patients or individuals without specifying ethnicity in general cohort.

5. Lines 38-40: “This is the first Russian report of the distribution of AR CAG repeats and the search of association between length of AR CAG repeat tract and impaired spermatogenesis in men from the general population” – the statement is not true, because these Russian studies have already been published, two of which ([52], [53]) are cited in the Discussion section:

1) Melikyan L.P., Bliznetz E.A., Polyakov A.V., Mironovich O.L., Kuznetsova I.A., Sorokina T.M., Shtaut M.I., Sedova A.O., Kurilo L.F., Solovova O.A., Chernykh V.B. Polymorphism of CAG repeats in exon 1 of the androgen receptor gene in Russian men with various forms of pathozoospermia. Russian Journal of Genetics. 2020;56(8): 1000-1005.

2) Melikyan L.P., Bliznetz E.A., Shtaut M.I., Sedova A.O., Sorokina T.M., Kurilo L.F., Polyakov A.V., Chernykh V.B. CAG polymorphism of the androgen receptor gene and semen parameters in pathozoospermic patients with and without Y chromosome microdeletions, and in normozoospermic men. Andrology and Genital Surgery. 2021;22(2):66-77. (In Russ.)

3) Melikyan L.P., Bliznetz E.A., Shtaut M.I., Sedova A.O., Sorokina T.M., Kurilo L.F., Polyakov A.V., Chernykh V.B. Study of the effect of CAG polymorphism of the androgen receptor (AR) gene on spermatological parameters in Russian men. Medical Genetics. 2020;19(10):19-31. (In Russ.)

4) Safina N.Yu., Yamandi T.A., Chernykh V.B., Akulenko L.V., Bogolyubov S.V., Vityazeva I.I., Ryzhkova O.P., Stepanova A.A., Adyan T.A., Bliznets E.A., Polyakov A.V. Genetic factors of male infertility, their combinations and the spermatological characteristics of men with fertility failures. Andrology and Genital Surgery. 2018;19(2):40-51. (In Russ.)

5) [52]

6) [53]

7) Mikhaylenko D.S., Babenko O.V., Kirillova E.A., Nikiforova O.K., Zaretskaya N.V., Kurnosova T.R. et al. Complex molecular genetic analysis of the AZF microdeletions, CFTR mutations, and CAG-repeat length of the AR gene in infertile men. Problems of Reproduction. 2005;11(6):52-5. (In Russ).

It is recommended to make this statement less categorical, as in the text of the article: this is not the first such study, but the first with so large cohort of Russian patients.

6. In my opinion, the Introduction contains many duplicated in meaning fragments: lines 47-51 and 78-79, lines 63-67 and 79-83, lines 66-77 and 83-90.

7. Fig. 1: the figure includes three separate diagrams of the distribution of the CAG repeat length under each other, although they have identical X/Y axes and the legend. It is recommended to put three diagrams into one graph, highlighting three different distributions with shading/color. It will help researchers to compare the distributions of allele lengths in the compared groups in the same coordinate system visually.

8. The content of table 1 completely repeats the content of figure 1 (in fact, it is a table for constructing diagrams of figure 1) and does not carry any new information. Such duplication of a table by a figure is redundant for a scientific publication. It is recommended to leave either Figure 1 or Table 1 in the article (or place the table in a supplementary file).

9. Table 5 and Figure 2: remarks are completely repeat the two previous paragraphs that refer to Table 1 and Figure 1.

10. Lines 473-474: it would be desirable to indicate which STS markers of the Y chromosome were tested to exclude microdeletions (or indicate the name of the test system/manufacturer).

11. Line 513: the manufacturer of the Nanophor-05 capillary genetic analyzer should be indicated.

12. Lines 532-544: the section 4.5. is about the formation of cohort from different ethnic groups – it is suitable to transfer and combine with section 4.1. at the beginning of the Materials and Methods section.

Author Response

Dear reviewer 1!

We thank and appreciate you for time and efforts spent on evaluating our manuscript. All your comments were helpful to us and, of course, improved our manuscript. We have followed all your recommendations, and the responses to your comments are set out below.

Sincerely yours,

Ludmila V. Osadchuk, DrSc, Professor

Federal Research Center the Institute of Cytology and Genetics, Siberian Branch of the Russian Academy of Sciences

Lavrentyev Ave. 10

630090, Novosibirsk, Russia

Answers to reviewer 1.

Comments and Suggestions for Authors

The study is dedicated to the actual problem of andrology - genetic basis of male infertility.

However, there are a number of comments to this article:

  1. Line 4, title: “...in Russian men from the general population” – if the samples are collected correctly, the words about the general population are unnecessary; if the samples are not representative, the study was not done So, it is recommended to remove the words about the general population. – Answer: The subjects of our study were young volunteers who were not selected for fertility or sperm parameters, so our study population was completely representative of the young part of the general population. Our participants were permanent residents of the cities, where the recruitment was performed, and have been exposed to the same environmental and socio-cultural factors. A standardized recruitment and phenotyping protocol and questionnaire were identical at all five cities to minimize between-city differences; the study groups were quite similar in social status. In these words (“the general population”), we emphasize that the study was conducted on volunteers who previously had no information about their fertility status, were not treated in reproductive clinics. In contrast, most studies on the AR CAG repeats are conducted on men with reproductive abnormalities, for example, with defects in spermatogenesis. Thus, we assume that the use of the term "general population" is adequate and informative in the text of the manuscript, but we have removed these words from the title.

  1. To the entire text: the name of the human AR gene should be written in italic in accordance with the international HUGO nomenclature. - Answer: We corrected our mistake in all places of our manuscript, but it should be noted that in some articles the AR gene was written in an ordinary font (not italic).

  1. Line 13-14: “…the androgen receptor (AR) that is assumed to negatively affect male fertility” – not the androgen receptor itself negatively affects fertility, but only some of its

- Answer: Done. “The search for the causes of male infertility allowed identifying a number of genetic factors including a single X-linked gene of the androgen receptor (AR), some of its alleles are assumed to negatively affect male fertility.”

  1. Line 18: “…young Russian men of different ethnicities…” – Russians cannot be of different nationalities; they are a separate ethnic group. Perhaps, the authors mean they collected material from patients (citizens of the Russian Federation) in 5 different regions with significantly different ethnic composition as described in the sample collection section. So, it is better to refer they as patients or individuals without specifying ethnicity in general cohort. - Answer: Russian as Russian citizen and Russian as nationality are translated into English in one word Russian. Anyway, you are right that for to define and separate their meanings we should use the term "citizen of Russia" to separate from Russian by nationality. As a result, we have removed the words “Russian men” from the title and text, replacing them with citizens of Russia where possible. I would like remind you that our study population was multi-ethnic and consisted of men of 20 ethnicities and descendants from ethnic mixed marriages.

  1. Lines 38-40: “This is the first Russian report of the distribution of AR CAG repeats and the search of association between length of AR CAG repeat tract and impaired spermatogenesis in men from the general population” – the statement is not true, because these Russian studies have already been published, two of which ([52], [53]) are cited in the Discussion section:

  • Melikyan L.P., Bliznetz E.A., Polyakov A.V., Mironovich O.L., Kuznetsova I.A., Sorokina M., Shtaut M.I., Sedova A.O., Kurilo L.F., Solovova O.A., Chernykh V.B.

Polymorphism of CAG repeats in exon 1 of the androgen receptor gene in Russian men with various forms of pathozoospermia. Russian Journal of Genetics. 2020;56(8): 1000-1005.

  • Melikyan L.P., Bliznetz E.A., Shtaut M.I., Sedova A.O., Sorokina T.M., Kurilo L.F., Polyakov A.V., Chernykh V.B. CAG polymorphism of the androgen receptor gene and semen parameters in pathozoospermic patients with and without Y chromosome microdeletions, and in normozoospermic Andrology and Genital Surgery. 2021;22(2):66-77. (In Russ.)

  • Melikyan L.P., Bliznetz E.A., Shtaut M.I., Sedova A.O., Sorokina T.M., Kurilo L.F., Polyakov V., Chernykh V.B. Study of the effect of CAG polymorphism of the androgen receptor (AR) gene on spermatological parameters in Russian men. Medical Genetics. 2020;19(10):19-31. (In Russ.)

  • Safina Yu., Yamandi T.A., Chernykh V.B., Akulenko L.V., Bogolyubov S.V., Vityazeva I.I., Ryzhkova O.P., Stepanova A.A., Adyan T.A., Bliznets E.A., Polyakov A.V. Genetic factors of male infertility, their combinations and the spermatological characteristics of men with fertility failures. Andrology and Genital Surgery. 2018;19(2):40-51. (In Russ.)

5) [52]

6) [53]

7) Mikhaylenko D.S., Babenko O.V., Kirillova E.A., Nikiforova O.K., Zaretskaya N.V., Kurnosova T.R. et al. Complex molecular genetic analysis of the AZF microdeletions, CFTR mutations, and CAG-repeat length of the AR gene in infertile men. Problems of Reproduction. 2005;11(6):52-5. (In Russ).

It is recommended to make this statement less categorical, as in the text of the article: this is not the first such study, but the first with so large cohort of Russian patients. - Answer: Here we kept in mind that our study is the first large Russian study carried out on men from the general population. Of course, we know about other Russian studies on the CAG polymorphism, but they were conducted on patients of reproductive clinics or centres, so the participants were preselected by fertility or semen quality. These studies have a big value for reproductive medicine, but in contrast to our study, they have different aims. These studies did not provide information about the variability of the length of CAG repeats of the AR gene and distributions of AR CAG repeat alleles in the general population and associations with semen parameters in different ethnic subgroups. Our population study based on random samples from the general population is more informative in this regard, since it more accurately reflects the situation in the population.

  1. In my opinion, the Introduction contains many duplicated in meaning fragments: lines 47-51 and 78-79, lines 63-67 and 79-83, lines 66-77 and 83-90. - Answer: Your question is not If we compare the content of fragments, for example, in lines 47-51 and 78-79, they contain completely different information. The same conclusion will be drawn in the other two fragments (lines 63-67 and 79-83, lines 66-77 and 83-90). Perhaps the numbering of the lines was broken, although there were no direct duplications. In any case, we have again looked through the entire manuscript and corrected all the duplicate fragments where possible.

  1. 1: the figure includes three separate diagrams of the distribution of the CAG repeat length under each other, although they have identical X/Y axes and the legend. It is recommended to put three diagrams into one graph, highlighting three different distributions with shading/color. It will help researchers to compare the distributions of allele lengths in the compared groups in the same coordinate system visually. - Answer: Done.

  1. The content of table 1 completely repeats the content of figure 1 (in fact, it is a table for constructing diagrams of figure 1) and does not carry any new information. Such duplication of a table by a figure is redundant for a scientific It is recommended to leave either Figure 1 or Table 1 in the article (or place the table in a supplementary file). - Answer: Done. We left Figure 1 and placed Table 1 in a supplementary file.

  1. Table 5 and Figure 2: remarks are completely repeat the two previous paragraphs that refer to Table 1 and Figure 1. - Answer: We left Figure 2 and placed Table 5 in a supplementary file.

  1. Lines 473-474: it would be desirable to indicate which STS markers of the Y chromosome were tested to exclude microdeletions (or indicate the name of the test system/manufacturer). - Answer: The evaluation principle of the Y-chromosome deletions is available in a supplementary file.

  1. Line 513: the manufacturer of the Nanophor-05 capillary genetic analyzer should be - Answer: Done. Genotyping of the AR CAG repeats was performed on a “Nanophor-05” sequencer (Syntol, Russia).

  1. Lines 532-544: the section 4.5. is about the formation of cohort from different ethnic groups – it is suitable to transfer and combine with section 4.1. at the beginning of the Materials and Methods section. - Answer:

Reviewer 2 Report

In this study, authors investigated the variability of the length of CAG repeats of the androgen receptor gene and possible associations of its genetic variants with semen quality and reproductive hormone levels in a Russian population. Authors also stratified the population in ethnic subgroups. The article is well-written and of interest. Similar studies have been carried out in different populations; as the ethnic and geography influences the allelic variants of the genes, it is interest to report results for a Russian population for the first time. Hence, I recommend publication after minor revisions:

Abstract: “….WAS recruited….” (line 19)

“ Male infertility is a multi-factorial and multi-genetic disorder and the prevalence of male 11 infertility in the world is estimated at 7.0%.” this sentence can be removed (the % does not match with the introduction – line 49)

Remove the references from the abstract.

For the sentence “ significant difference was found in the frequency 26 distribution and the mean values for the CAG repeat length between group with normal (23.2±3.3) 27 and impaired semen quality (23.9±3.2).” please, include the p value.

There is no mention to the hormone results in the abstract.

Line 69: write “six” as “6”.

Please doublecheck refs 8 and 9 for line 70, if they really investigate ethnic differences.

Lines 71-74: here authors cite just 1 study. It is better to start the sentence as “ According to Osadchuk,and Osadchuk, in males of the Negroid race….”

Check the abbreviations for androgen receptor throughout the entire manuscript.

Line 79: please delete “and is the most common aetiology”.

For the sentence “Most of the studies have shown an association of the expanded CAG repeats with male infertility and an impairment of spermatogenesis, although this was not true for all studies [6; 14; 15]” please cite in different parenthesis the studies were an impairment was observed or not.

Line 88: WAS not observed.

Line 89: “although this pattern was not absolute” is unclear. Please rephrase.

In figure 1, the number 22 should be under the highest bar.

Line 126: please include the p value.

Table 1 is difficult to read and not really informative. Maybe to be moved to supplementary material? Same table 5.

Are you sure that height differs between normal and impaired semen quality groups? The values look almost the same.

Line 189 is part of the results. There should not be a reference.

Please discuss the limitations of this study.

Author Response

Dear reviewer 2!

We thank and appreciate you for time and efforts spent on evaluating our manuscript. All your comments were helpful to us and, of course, improved our manuscript. We have followed all your recommendations, and the responses to your comments are set out below.

Sincerely yours,

Ludmila V. Osadchuk, DrSc, Professor

Federal Research Center the Institute of Cytology and Genetics, Siberian Branch of the Russian Academy of Sciences

Lavrentyev Ave. 10

630090, Novosibirsk, Russia

Answers to Editor-in-Chief and our reviewers.

Dear colleagues!

We thank and appreciate you and our reviewers for time and efforts spent on evaluating our manuscript. All your comments were helpful to us and, of course, improved our manuscript. We have followed all your recommendations, and the responses to your comments are set out below.

Sincerely yours,

Ludmila V. Osadchuk, DrSc, Professor

Federal Research Center the Institute of Cytology and Genetics, Siberian Branch of the Russian Academy of Sciences

Lavrentyev Ave. 10

630090, Novosibirsk, Russia

Answers to reviewer 1.

Comments and Suggestions for Authors

The study is dedicated to the actual problem of andrology - genetic basis of male infertility. However, there are a number of comments to this article:

  1. Line 4, title: “...in Russian men from the general population” – if the samples are collected correctly, the words about the general population are unnecessary; if the samples are not representative, the study was not done correctly. So, it is recommended to remove the words about the general population. – Answer: The subjects of our study were young volunteers who were not selected for fertility or sperm parameters, so our study population was completely representative of the young part of the general population. Our participants were permanent residents of the cities, where the recruitment was performed, and have been exposed to the same environmental and socio-cultural factors. A standardized recruitment and phenotyping protocol and questionnaire were identical at all five cities to minimize between-city differences; the study groups were quite similar in social status. In these words (“the general population”), we emphasize that the study was conducted on volunteers who previously had no information about their fertility status, were not treated in reproductive clinics. In contrast, most studies on the AR CAG repeats are conducted on men with reproductive abnormalities, for example, with defects in spermatogenesis. Thus, we assume that the use of the term "general population" is adequate and informative in the text of the manuscript, but we have removed these words from the title.

  1. To the entire text: the name of the human AR gene should be written in italic in accordance with the international HUGO nomenclature. - Answer: Done. We corrected our mistake in all places of our manuscript, but it should be noted that in some articles the AR gene was written in an ordinary font (not italic).

  1. Line 13-14: “…the androgen receptor (AR) that is assumed to negatively affect male fertility” – not the androgen receptor itself negatively affects fertility, but only some of its alleles. - Answer: Done. “The search for the causes of male infertility allowed identifying a number of genetic factors including a single X-linked gene of the androgen receptor (AR), some of its alleles are assumed to negatively affect male fertility.”

  1. Line 18: “…young Russian men of different ethnicities…” – Russians cannot be of different nationalities; they are a separate ethnic group. Perhaps, the authors mean they collected material from patients (citizens of the Russian Federation) in 5 different regions with significantly different ethnic composition as described in the sample collection section. So, it is better to refer they as patients or individuals without specifying ethnicity in general cohort. - Answer: Done. Russian as Russian citizen and Russian as nationality are translated into English in one word Russian. Anyway, you are right that for to define and separate their meanings we should use the term "citizen of Russia" to separate from Russian by nationality. As a result, we have removed the words “Russian men” from the title and text, replacing them with citizens of Russia where possible. I would like remind you that our study population was multi-ethnic and consisted of men of 20 ethnicities and descendants from ethnic mixed marriages.

  1. Lines 38-40: “This is the first Russian report of the distribution of AR CAG repeats and the search of association between length of AR CAG repeat tract and impaired spermatogenesis in men from the general population” – the statement is not true, because these Russian studies have already been published, two of which ([52], [53]) are cited in the Discussion section:

1) Melikyan L.P., Bliznetz E.A., Polyakov A.V., Mironovich O.L., Kuznetsova I.A., Sorokina T.M., Shtaut M.I., Sedova A.O., Kurilo L.F., Solovova O.A., Chernykh V.B. Polymorphism of CAG repeats in exon 1 of the androgen receptor gene in Russian men with various forms of pathozoospermia. Russian Journal of Genetics. 2020;56(8): 1000-1005.

2) Melikyan L.P., Bliznetz E.A., Shtaut M.I., Sedova A.O., Sorokina T.M., Kurilo L.F., Polyakov A.V., Chernykh V.B. CAG polymorphism of the androgen receptor gene and semen parameters in pathozoospermic patients with and without Y chromosome microdeletions, and in normozoospermic men. Andrology and Genital Surgery. 2021;22(2):66-77. (In Russ.)

3) Melikyan L.P., Bliznetz E.A., Shtaut M.I., Sedova A.O., Sorokina T.M., Kurilo L.F., Polyakov A.V., Chernykh V.B. Study of the effect of CAG polymorphism of the androgen receptor (AR) gene on spermatological parameters in Russian men. Medical Genetics. 2020;19(10):19-31. (In Russ.)

4) Safina N.Yu., Yamandi T.A., Chernykh V.B., Akulenko L.V., Bogolyubov S.V., Vityazeva I.I., Ryzhkova O.P., Stepanova A.A., Adyan T.A., Bliznets E.A., Polyakov A.V. Genetic factors of male infertility, their combinations and the spermatological characteristics of men with fertility failures. Andrology and Genital Surgery. 2018;19(2):40-51. (In Russ.)

5) [52]

6) [53]

7) Mikhaylenko D.S., Babenko O.V., Kirillova E.A., Nikiforova O.K., Zaretskaya N.V., Kurnosova T.R. et al. Complex molecular genetic analysis of the AZF microdeletions, CFTR mutations, and CAG-repeat length of the AR gene in infertile men. Problems of Reproduction. 2005;11(6):52-5. (In Russ).

It is recommended to make this statement less categorical, as in the text of the article: this is not the first such study, but the first with so large cohort of Russian patients. - Answer: Here we kept in mind that our study is the first large Russian study carried out on men from the general population. Of course, we know about other Russian studies on the CAG polymorphism, but they were conducted on patients of reproductive clinics or centres, so the participants were preselected by fertility or semen quality. These studies have a big value for reproductive medicine, but in contrast to our study, they have different aims. These studies did not provide information about the variability of the length of CAG repeats of the AR gene and distributions of AR CAG repeat alleles in the general population and associations with semen parameters in different ethnic subgroups. Our population study based on random samples from the general population is more informative in this regard, since it more accurately reflects the situation in the population.

  1. In my opinion, the Introduction contains many duplicated in meaning fragments: lines 47-51 and 78-79, lines 63-67 and 79-83, lines 66-77 and 83-90. - Answer: Your question is not clear. If we compare the content of fragments, for example, in lines 47-51 and 78-79, they contain completely different information. The same conclusion will be drawn in the other two fragments (lines 63-67 and 79-83, lines 66-77 and 83-90). Perhaps the numbering of the lines was broken, although there were no direct duplications. In any case, we have again looked through the entire manuscript and corrected all the duplicate fragments where possible.

  1. Fig. 1: the figure includes three separate diagrams of the distribution of the CAG repeat length under each other, although they have identical X/Y axes and the legend. It is recommended to put three diagrams into one graph, highlighting three different distributions with shading/color. It will help researchers to compare the distributions of allele lengths in the compared groups in the same coordinate system visually. - Answer: Done.

  1. The content of table 1 completely repeats the content of figure 1 (in fact, it is a table for constructing diagrams of figure 1) and does not carry any new information. Such duplication of a table by a figure is redundant for a scientific publication. It is recommended to leave either Figure 1 or Table 1 in the article (or place the table in a supplementary file). - Answer: Done. We left Figure 1 and placed Table 1 in a supplementary file.

  1. Table 5 and Figure 2: remarks are completely repeat the two previous paragraphs that refer to Table 1 and Figure 1. - Answer: Done. We left Figure 2 and placed Table 5 in a supplementary file.

  1. Lines 473-474: it would be desirable to indicate which STS markers of the Y chromosome were tested to exclude microdeletions (or indicate the name of the test system/manufacturer). - Answer: Done. The evaluation principle of the Y-chromosome deletions is available in a supplementary file.

  1. Line 513: the manufacturer of the Nanophor-05 capillary genetic analyzer should be indicated. - Answer: Done. Genotyping of the AR CAG repeats was performed on a “Nanophor-05” sequencer (Syntol, Russia).

  1. Lines 532-544: the section 4.5. is about the formation of cohort from different ethnic groups – it is suitable to transfer and combine with section 4.1. at the beginning of the Materials and Methods section. - Answer: Done.

Answers to reviewer 2.

Comments and Suggestions for Authors

In this study, authors investigated the variability of the length of CAG repeats of the androgen receptor gene and possible associations of its genetic variants with semen quality and reproductive hormone levels in a Russian population. Authors also stratified the population in ethnic subgroups. The article is well-written and of interest. Similar studies have been carried out in different populations; as the ethnic and geography influences the allelic variants of the genes, it is interest to report results for a Russian population for the first time. Hence, I recommend publication after minor revisions:

Abstract: “….WAS recruited….” (line 19) - Answer: Done.

“ Male infertility is a multi-factorial and multi-genetic disorder and the prevalence of male 11 infertility in the world is estimated at 7.0%.” this sentence can be removed (the % does not match with the introduction – line 49) – Answer: Done. Indeed, estimates of the prevalence of male infertility vary greatly. We have made corrections to the abstract so that the values in the abstract and in the text do not differ.

Remove the references from the abstract. - Answer: Sorry, but there are no references in the abstract.

For the sentence “ significant difference was found in the frequency 26 distribution and the mean values for the CAG repeat length between group with normal (23.2±3.3) 27 and impaired semen quality (23.9±3.2).” please, include the p value. – Answer: Done. We have included (p≤0.05) in this sentence.

There is no mention to the hormone results in the abstract. – Answer: Done. We have included (line 29-30) “however, hormonal parameters did not differ between the long and short CAG categories, with the exception of estradiol” and (line 37) “Hormonal parameters did not differ between three CAG repeat categories in men of all ethnic groups”. We also removed the words “reproductive hormonal status" from the title due to the lack of hormonal differences.

Line 69: write “six” as “6”. – Answer: Done. We replaced “six” with “6”.

Please doublecheck refs 8 and 9 for line 70, if they really investigate ethnic differences. - Answer: Done. Refs 8 and 9 investigated ethnic differences. Ref 8 (Irvine et al., 1995) studied the prevalence of CAG repeat length in African-American, non-Hispanic whites, and in Asian males; refs 9 (Sartor er al., 1999) investigated non-Hispanic white and black men.

Lines 71-74: here authors cite just 1 study. It is better to start the sentence as “ According to Osadchuk,and Osadchuk, in males of the Negroid race….” – Answer: Done. We replaced “Briefly…” with “A review of the available studies showed that….”

Check the abbreviations for androgen receptor throughout the entire manuscript. - Answer: Done. We have corrected the AR gene (written in italics) throughout the text.

Line 79: please delete “and is the most common aetiology”. – Answer: Done. We have deleted this.

For the sentence “Most of the studies have shown an association of the expanded CAG repeats with male infertility and an impairment of spermatogenesis, although this was not true for all studies [6; 14; 15]” please cite in different parenthesis the studies were an impairment was observed or not. – Answer: Here we cite 3 reviews, each of which presents alternative results on the relationship between the CAG repeat length and male fertility, so they cannot be separated.

Line 88: WAS not observed. - Answer: We used a phrase ”such relationships were not observed in some…..”.

Line 89: “although this pattern was not absolute” is unclear. Please rephrase. – Answer: Done. We have deleted this unclear phrase and changed the sentence as follows “…but such relationships were occasionally observed in some Asian and African populations”.

In figure 1, the number 22 should be under the highest bar. - Answer: Done. We have corrected all figures.

Line 126: please include the p value. – Answer: Done. We have included the p value (p≤0.05).

Table 1 is difficult to read and not really informative. Maybe to be moved to supplementary material? Same table 5. – Answer: We have placed Table 1 and 5 in a supplementary file.

Are you sure that height differs between normal and impaired semen quality groups? The values look almost the same. – Answer: We are confident in the difference in growth, although the difference is weak, it is statistically significant (p=0.024).

Line 189 is part of the results. There should not be a reference. - Answer: Done. We have moved that fragment to the section “Materials and Methods”.

Please discuss the limitations of this study. - Answer: Done. The limitations of the study were included into section Discussion: “In our study, there are few limitations associated with effects of AR CAG repeat alleles on semen parameters. Firstly, more research is needed to evaluate the effects of CAG repeats in Yakut ethnic group to confirm or reject our pilot observations. Our multi-ethnic research may also serve as a starting point for further studies, taken into account a wide diversity of nationalities in Russia. Secondly, some studies suggest that GGN repeats in the AR gene or specific combinations of CAG and GGN triplets can modulate the AR function [30, 39, 40]. Additional studies are required to identify a role of independent or combinatory effects of CAG and GGN repeat lengths in spermatogenesis of men from Russia. Thirdly, it is assumed that long-term research projects will be aimed at whole-exome sequencing in representatives of various ethnic groups from Russia, providing a search of pathogenic ethno-specific genetic variants, significant associations of molecular genetic markers with the parameters of impaired spermatogenesis and new genetic loci involved in the control of spermatogenesis”.

Round 2

Reviewer 1 Report

The authors took into account all the comments of the reviewer. This corrected manuscript could be approved for publication.